# De-Noising of Magnetotelluric Signals by Discrete Wavelet Transform and SVD Decomposition

**Rui Zhou [1], Jiangtao Han [1,2,3,*], Zhenyu Guo [1] and Tonglin Li [1,2,3]**

1. College of Geoexploration Science and Technology, Jilin University, Changchun 130026, China; zhour21@mails.jlu.edu.cn (R.Z.); zhenyug20@mails.jlu.edu.cn (Z.G.); litl@jlu.edu.cn (T.L.)
2. Key Laboratory of Applied Geophysics, Ministry of Natural Resources, Changchun 130026, China
3. Changbai Mountain Volcano Comprehensive Geophysics Field Science Observation and Research Station of Ministry of Education, Changchun 130026, China
* Correspondence: hanjt@jlu.edu.cn

**Abstract:** Magnetotelluric (MT) sounding data can easily be damaged by various types of noise, especially in industrial areas, where the quality of measured data is poor. Most traditional de-noising methods are ineffective to the low signal-to-noise ratio of data. To solve the above problem, we propose the use of a de-noising method for the detection of noise in MT data based on discrete wavelet transform and singular value decomposition (SVD), with multiscale dispersion entropy and phase space reconstruction carried out for pretreatment. No "over processing" takes place in the proposed method. Compared with wavelet transform and SVD decomposition in synthetic tests, the proposed method removes the profile of noise more completely, including large-scale noise and impulse noise. For high levels or low levels of noise, the proposed method can increase the signal-to-noise ratio of data more obviously. Moreover, application to the field MT data can prove the performance of the proposed method. The proposed method is a feasible method for the elimination of various noise types and can improve MT data with high noise levels, obtaining a recovery in the response. It can improve abrupt points and distortion in MT response curves more effectively than the robust method can.

**Keywords:** magnetotelluric; multiscale dispersion entropy; phase space reconstruction; wavelet transform; SVD decomposition

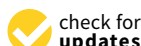



## 1. Introduction

Magnetotelluric (MT) sounding can be used to explore subsurface structures by observing the natural orthogonal electric and magnetic fields. Compared with controlled-source signals, natural MT signals have the advantages of a wide frequency range, deep exploration depth, and easy-to-use equipment. Therefore, MT is widely used in many fields, such as marine exploration, volcano monitoring, and geodynamic interpretation [1–3]. However, during the period of observation, MT data are vulnerable to distortions from electromagnetic wave emitted by high-voltage lines, communication radio stations, and underground mining machines around monitoring sites. In general, the intensity of noise is stronger than that of the MT signal, which can even be completely submerged by cultural noise, causing isolated points and distortions in apparent resistivity-phase curves, which are used to image geoelectrical structures. Therefore, it is necessary to improve the continuity and smoothness of the apparent resistivity-phase curves calculated based on impedance estimation. Two major types of methods are used in the impedance estimation of MT data according to the processing objects: frequency-domain methods and time-domain methods. One is an electric and magnetic field spectrum or power spectrum obtained by time–frequency transform, while the other is the time series of electric and magnetic fields and is directly observed.

Least squares estimation [4–6] and the robust method [7–9] are classic frequency-domain de-noising methods used in MT. There is a high correlation between horizontal MT signal components; thus, the least squares method, which uses a cross power spectrum, can eliminate the uncorrelated noise when there is no correlation between the signal and noise. However, the noise is always correlated with measured data; thus, the improvement of the traditional least squares method is biased. Based on the least squares method, the robust method was introduced for MT de-noising and can reduce the weight coefficients of noise through statistical analysis and coherency [7,10]. Smirnov (2003) proposed the criteria of the weight coefficient based on repeated median (RM), which is suitable for use in a strong-noise environment [11]. Chave (2014) introduced the maximum likelihood estimation method, which makes use of the distribution of noise data to calculate weights [12]. However, the robust method is insensitive to correlated noise and dependent on the quality of the data. In order to reduce noise weight coefficients, the amplitude of the signal must be far higher than that of the noise [13,14]. For correlated noise, Gamble et al. (1979) proposed the remote reference (RR) method. However, in industrialized areas such as ore mining areas, the quality of data is poor, meaning that any improvement made by the RR method is weak [15–17]. Oettinger et al. (2001) further introduced a signal-to-noise separation method with two remote reference sites to improve the de-noising ability of the RR method when used for a single site [18]. However, there has always been no solution to selecting the location of reference sites and the method is expensive.

In addition to the above-mentioned frequency-domain methods, time-domain methods can be used to improve the accuracy of impedance estimation. Empirical mode decomposition (EMD) [19,20] adaptively decomposes signals to intrinsic mode functions (IMFs) with different frequencies and a residual component. However, there the mode-mixing phenomenon occurs in EMD. The use of EEMD [21] and CEEMD [22] has been further proposed, which weakens the effect of mode-mixing, but there are also other problems, such as the existence of recovery error and the longer computing times needed. The mathematical morphological filtering method [23] and sparse decomposition algorithm [24] are effective only when the selection of parameters in the method is appropriate, which is affected by human factors that interfere greatly, and it is difficult to suppress multiple types of noise simultaneously. Morlet et al. (1982) introduced the method of continuous wavelet transform to signal analysis [25] and a series of mother wavelet transforms was further proposed [26–29]. Kumar and Foufoula-Georgiou (1997) proposed the application of discrete wavelet transform to geophysical signal processing [30]. Trad and Travassos (2000) introduced the wavelet threshold de-noising method based on the distribution of noise data [31]. Escalas et al. (2013) introduced polarization analysis for processing signals in wavelet domain in order to separate signals and noise [32]. However, the improvement of wavelet transform is also dependent on the selection of the parameters, such as the mother wavelet function and wavelet decomposition levels. Meanwhile, the most common way to select parameters for a method is through experience or repeated experiments, but this approach is influenced by subjective factors as well as being time-consuming.

Obviously, these methods are ineffective in improving the quality of data when the signal-to-noise ratio is low and the selection of the parameters used in methods is inappropriate. When the selected parameters are inappropriate, losses of signals can occur, or the methods may not even be capable of removing noise. Thus, we propose the use of a de-noising method based on discrete wavelet transform and iterative singular value decomposition (SVD). In this paper, the traditional SVD is improved by iterative loop that can extract noise more completely. Combined with discrete wavelet transform, the use of this method for the identification of various types of noise, including large-scale noise and impulse noise, is feasible. Meanwhile, multiscale dispersion entropy and phase space reconstruction are applied in this method to reduce the loss of signal and adaptively select parameters. To demonstrate the accuracy and stability of the proposed method, the proposed method is applied to process the MT field data of Linze area and Qilian area, China. Compared with traditional SVD decomposition, wavelet transform and the robust method,

the proposed method can remove the various noise more thoroughly and obtain the useful MT response curves, which more truly reflect the subsurface electromagnetic structure.

## 2. Methods

### 2.1. Multiscale Dispersion Entropy

In general, compared with natural MT signals, cultural noise is expected to show a certain direction of polarization. Multiscale dispersion entropy (*MDE*) can identify the differences between the signal and cultural noise, and the *MDE* of cultural noise is smaller than that of signals. Compared with multiscale sample entropy (MSE), multiscale fuzzy entropy (MFE), and multiscale approximate entropy (MAE), the calculation of *MDE* is simpler and faster [33]. Multiscale dispersion entropy is defined as follows [34,35]:

(1) Time series $u_j$ ($j$ = 1, 2, ... , $N$) are taken via a coarse graining preprocessing for non-overlapping segments $x_b$ and are obtained as follows:

$$x_b = \frac{1}{\tau} \sum_{j=(b-1)*\tau+1}^{b*\tau} u_j, (1 \leq b \leq L) \tag{1}$$

where $\tau$ is the scale factor, $L = [N/\tau]$ represents the length of the data segment, $N$ is the length of time series, and [●] is the integral function.

(2) $x_b$ is mapped to $c$ classes from 1 to $c$. First, the normal cumulative distribution function (NCDF) is employed to map the segment to $y_b$ from 0 to 1:

$$y_b = \frac{1}{\sigma\sqrt{2\pi}} \int_{-\infty}^{x_b} e^{\frac{-(t-rms)^2}{2\sigma^2}} dt \tag{2}$$

where *rms* represents the root mean square and $\sigma$ is the standard deviation (SD) of the segment, $t$ is the length of observation time in each data segment. A linear algorithm $z_b^c = round(c \bullet y_b + 0.5)$ is used for $y_b$ to $z_b^c$ from 1 to $c$. Then, the embedding dimension $m$ and time delay $d$ are introduced to reconstruct $z_i^{m,c}$ as follows:

$$z_i^{m,c} = \left\{ z_i^c, z_{i+d}^c \cdots z_{i+(m-1)d}^c \right\} \tag{3}$$

where $i$ = 1, 2 ... $L-(m-1)d$. The number of possible dispersion patterns $\pi_{v_0 \cdots v_{m-1}}$ that can be assigned to each time-series $z_i^{m,c}$ is equal to $c^m$, since the signal has $m$ members and each member can be an integer from 1 to c. Finally, for each $c^m$ potential dispersion pattern, the relative frequency is defined as follows:

$$p\left(\pi_{v_0 \cdots v_{m-1}}\right) = \frac{Number\{i | i \leq L - (m-1)d, \ z_i^{m,c} \ \ has \ \ type \ \ \pi_{v_0 \cdots v_{m-1}} \}}{L - (m-1)d} \tag{4}$$

where $p\left(\pi_{v_0 \cdots v_{m-1}}\right)$ represents the number of dispersion patterns $\pi_{v_0 \cdots v_{m-1}}$ that are assigned to $z_i^{m,c}$ divided by the total number of embedding signals with embedding dimension $m$. $v_0 = z_i^c$, $v_1 = z_{i+d}^c \cdots$ , $v_{m-1} = z_{i+(m-1)d}^c$.

(3) The *MDE* is obtained as follows:

$$MDE_{rms} = E(x, \tau, m, c, d) = [e_1, e_2 \cdots, e_\tau] \tag{5}$$

where $e(x, m, c, d) = -\sum_{\pi=1}^{c^m} p\left(\pi_{v_0 \cdots v_{m-1}}\right) \times ln\left(p\left(\pi_{v_0 \cdots v_{m-1}}\right)\right)$ represents the dispersion entropy based on Shannon's entropy.

### 2.2. Phase Space Reconstruction

As a part of the chaos theory, phase space reconstruction is applied to indicate the number of real phase spaces in one-dimensional vectors, such as dynamical systems and

time series, with differences seen between the correlation of data [36–38]. The algorithm is employed twice in the proposed method for (1) the selection of wavelet decomposition levels and (2) matrix construction before SVD decomposition.

Suppose $U = \{u_1, u_2 \dots u_N\}$, where $N$ is the length of the signal, is reconstructed to different phase space vectors $x_i = \{u_i, u_{i+\tau} \dots u_{i\ +(m-1)\tau}\}$ ($i = 1, 2, \dots, m$) through the method of delay, where $\tau$ denotes time delay and $m$ is the embedding dimension. Thus, the two parameters are necessary for the reconstruction. In general, various methods can be used to obtain $\tau$, such as autocorrelation function [39], experience [40], and mutual information function [41]. However, compared with the above two methods, the third method is expected to process non-linear and non-stationary data. For the selection of $m$ in this paper, false nearest neighbors (FNN) [42] are introduced to obtain $m$, where $m$ must satisfy $m > 2h + 1$ (Takens theory) [43] and $h$ represents the real dimension of attractors.

### 2.2.1. Mutual Information Function

Consider a general coupled system $(S,Q)$, where $[s, q] = [x(t), x(t + \tau)]$ and the entropy of the system is $H$. When $x$ is measured at time $t$, the average uncertainty of $x$ at time $t + \tau$ is defined by averaging $H(Q \mid s_i)$ over $s_i$ as follows:

$$\begin{aligned} H(Q|S) \ &= \sum_i P_s(s_i) H(Q|s_i) \\ &= H(S,Q) - H(S) \end{aligned} \tag{6}$$

where $H(Q \mid s)$ is the uncertainty of $q$ given a measurement of $s$ and $H(S,Q)$ is the uncertainty of $q$ in isolation. Thus, the measurement of $s$ that reduces the uncertainty of $q$ is:

$$\begin{aligned} I(Q,S) \ &= H(Q) + H(S) - H(S,Q) \\ &= I(S,Q) \end{aligned} \tag{7}$$

where $I(S,Q)$ is the mutual information of coupled system $(S,Q)$. Thus, mutual information can be generalized as follows:

$$I_n(X_0, X_1, \cdots, X_n) = \sum_j \big(H(X_j)\big) - H(X_0, X_1, \cdots, X_n) \tag{8}$$

when the one-dimensional vector is the time-delay reconstruction and the first minimum in $I_n(\tau)$ is employed as time delay $\tau$.

### 2.2.2. False Nearest Neighbors

The nearest neighbor of $x_i = \{u_i, u_{i+\tau} \dots u_{i+(m-1)\tau}\}$ by $x_i^0 = \{u_i^0, u_{i+\tau}^0 \dots u_{i+(m-1)\tau}^0\}$ within a certain distance in $m$ dimensions is denoted. Then, the square Euclidian distance is as follows:

$$R_i(m) = \sqrt{\sum_{j=0}^{m-1} \big(u_{i+j} - u^0{}_{i+j}\big)^2} \tag{9}$$

When dimension $m$ increases to dimension $m + 1$, the distance is expressed as:

$$R_i(m+1) = \sqrt{\sum_{j=0}^{m-1} \big(u_{i+j} - u^0{}_{i+j}\big)^2 + \big(u_{i+m} - u^0{}_{i+m}\big)^2} \tag{10}$$

Then, $a_1(i, m)$ is given as follows:

$$a_1(i, m) = \frac{R_i(m+1)}{R_i(m)} \tag{11}$$

For $a_1(i, m) \geq 10$, the neighbor is identified as false; in contrast, $m$ is the embedding dimension.

### 2.3. Discrete Wavelet Transform

Wavelet transform is defined as continuous wavelet transform (CWT) [44] and discrete wavelet transform (DWT) [30]. DWT has been introduced in the proposed method, in order to inversely transform the signal in the wavelet domain into a time series, which is available for further processing by SVD decomposition. Mother wavelet function and wavelet decomposition levels are the parameters used in wavelet transform. In MT signals, coif$N$ is most suitable as the mother wavelet function, where $N = 5$ obtains a better result [45,46]. Meanwhile, the wavelet decomposition level in the proposed method is calculated using phase space reconstruction.

The definition of the DWT of $u(t)$ is:

$$W_f(m, n) = \lambda_0^{-m/2} \int u(t)\psi\left(\lambda_0^{-m}t - nt_0\right)dt \tag{12}$$

$$\Psi_{m,n}(t) = \frac{1}{\sqrt{\lambda_0^m}}\Psi\left(\frac{t - nt_0\lambda_0^m}{\lambda_0^m}\right) = \lambda_0^{-m/2}\Psi\left(\lambda_0^{-m}t - nt\right) \tag{13}$$

Equation (13) represents the mother wavelet function. $m$ and $n$ are the integers. $\lambda_0$ is the scale. $t$ is the temporal shift and $t_0$ is initial condition. $W_f(m, n)$ denotes the wave coefficients, which satisfy the following expression:

$$A\|f\|^2 \leq \sum_m \sum_n \left|W_f(m, n)\right|^2 \leq B\|f\|^2 \tag{14}$$

where $A > 0$, $B < \infty$ are both characteristic constants of the wavelet and the choices of $\lambda_0$ and $t_0$ are correlated with them [30]. Thus, the inverse wavelet transform is obtained as follows:

$$f(t) = \frac{2}{A + B}\sum_m \sum_n W_f(m, n)\Psi_{m,n} + \gamma \tag{15}$$

### 2.4. SVD Decomposition

Suppose that $X = [x_1^T, x_2^T, \dots x_n^T]^T$, where $X \in R^{m \times n}$. The following formulation of the SVD method [47] is:

$$X = USV^T \tag{16}$$

where $S = \begin{cases} (diag(\sigma_1, \sigma_2, \cdots \sigma_P), O)^T, m > n \\ (diag(\sigma_1, \sigma_2, \cdots \sigma_P), O), m < n \end{cases}$, $S \in R^{m \times n}$ represents a diagonal matrix, $O$ is the zero matrix, $\sigma_1 \geq \sigma_2 \geq \dots \geq \sigma_p \geq 0$, and $p = \min(m, n)$, which are the singular values of $X$ in descending order. $U$ is an $m \times m$ matrix, of which the columns are orthonormal. $V$ is an $n \times n$ matrix, of which the rows are orthonormal.

In general, there are two methods for transforming a one-dimensional vector to a matrix [48]: one involves non-overlapping the signal to $N$ segments, while the other involves transforming the vector to a HERMIT matrix, where $\tau$ is 1. However, the methods above are special cases in the construction of the matrix and will not eliminate cultural noise effectively. Thus, the method of delay is applied in this paper; this selects the parameters $\tau$ and $m$ with the consideration of data in different actual situations. Compared with artificially selected parameters, this method improves the precision and time-consuming characteristics. Meanwhile, we improve SVD decomposition with the iteration until the *MDE* of all components is more than the threshold, which is the termination criterion for the iteration, to achieve the complete separation of signal and noise.

$$\sigma^i = \sigma_s^i + \sigma_n^i (i = 1, 2, \cdots, k) \tag{17}$$

$$Y = X_s + X_n = \begin{bmatrix} U_s & U_n \end{bmatrix}\begin{bmatrix} S_s & 0 \\ 0 & S_n \end{bmatrix}\begin{bmatrix} V_s^T \\ V_n^T \end{bmatrix} \tag{18}$$

where $\sigma_s{}^i$ and $X_s$ represent the effective signal section and $\sigma_n{}^i$ and $X_n$ are the noise section. The two-norm $f = \left(\|\sum \overline{X_s}\|_2\right)_{min}$ is employed to estimate the signal and the reconstruction matrix of the signal is expressed as follows:

$$\overline{X_s} = U_s S_s V_s{}^T \tag{19}$$

where $\overline{X_s}$ is the average of all the reconstruction matrices. Equations (15)–(18) are repeated until $MDE\left(\overline{X_S}\right) > \varepsilon$, where $\varepsilon$ is the threshold, for which the value is the same as the $MDE$ in Section 2.1, which is employed to filter noisy data.

The steps adopted in this paper are as follows (in Figure 1):

1.  Overlap data to segments to ensure the continuity of the data;
2.  Calculate the $MDE$ for each data segment, for which the value below the threshold is the noisy data segment;
3.  Apply phase space reconstruction to calculate the number of wavelet decomposition level in noisy data segments;
4.  Perform discrete wavelet transform and discrete wavelet inverse transform for multiple components;
5.  Decompose the components using iterative SVD to obtain the de-noised section;
6.  Reconstruct the MT de-noising signal with the useful data segments given in Step 1 and the de-noising data segments given in Step 5, where the average value of the two segments is adopted for the overlapping data.

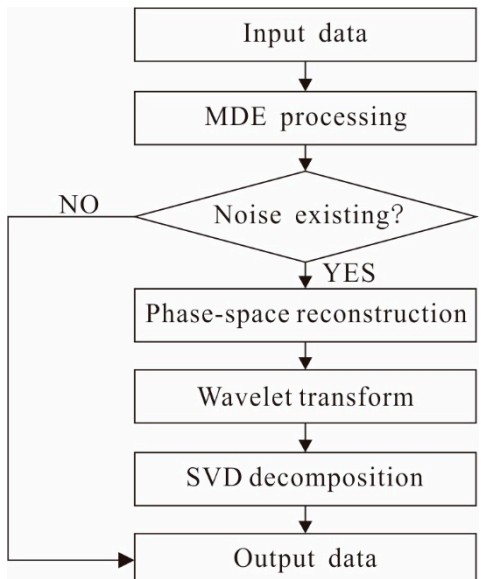

**Figure 1.** Steps of the proposed processing scheme.

### 3. Synthetic Cases

To test the effectiveness of the proposed method, various simulated noise was randomly added into natural MT signals with a high signal-to-noise ratio, such as square wave noise, charge–discharge triangular wave noise, and impulse noise. In order to compare the recovery results of different methods, we introduced signal-to-noise ratio (SNR), mean square error (MSE), normalized cross coherence (NCC), and data error (E) to evaluate the de-noising signals [49,50]:

$$SNR = 20lg\frac{\|y(n)\|_2}{\|y(n) - x(n)\|_2} \tag{20}$$

$$MSE = \sqrt{\frac{\sum_{n-1}^{N}(y(n) - x(n))^2}{N}} \tag{21}$$

$$\text{NCC} = \frac{\sum_{n-1}^{N} y(n) \cdot x(n)}{\sqrt{\left(\sum_{n-1}^{N} y^2(n)\right) \cdot \left(\sum_{n-1}^{N} x^2(n)\right)}} \tag{22}$$

$$\text{E} = \frac{\|y(n) - x(n)\|_2}{\|y(n)\|_2} \tag{23}$$

$$\text{S/N} = 20 lg \frac{\|y(n)\|_2}{\|x(n)\|_2} \tag{24}$$

where $y(n)$ represents the de-noised signal, $x(n)$ denotes the original signal, $\|\bullet\|_2$ is a 2-norm function, and S/N denotes the level of added noise. The value of 0 dB indicates that the intensity of the added noise is much stronger than the useful signal, which means that the noise level is high. On the contrary, 40 dB represents that the intensity of added noise is low. When S/N is 40 dB, the intensity of the noise is only one percent of the useful signal, meaning that it has little influence on the useful signal.

As shown in Figure 2, various types of noise (S/N = 0) are added into the synthetic noisy signal. In Figure 2a, the natural MT signal is measured in Linze Province with a sampling rate of 15 Hz. Figure 2b shows the addition of square wave noise to the time series, Figure 2c denotes the addition of charge–discharge triangular wave noise, Figure 2d shows the addition of impulse noise, and Figure 2e shows the addition of the above various types of noise. All the noise is added in random positions. Obviously, the useful signal is completely submerged and its amplitude is far less than that of the noise. Figure 2f–j show the frequency spectrum of Figure 2a–e. The amplitude shown in Figure 2f is less than the amplitude shown in Figure 2g,h,j.

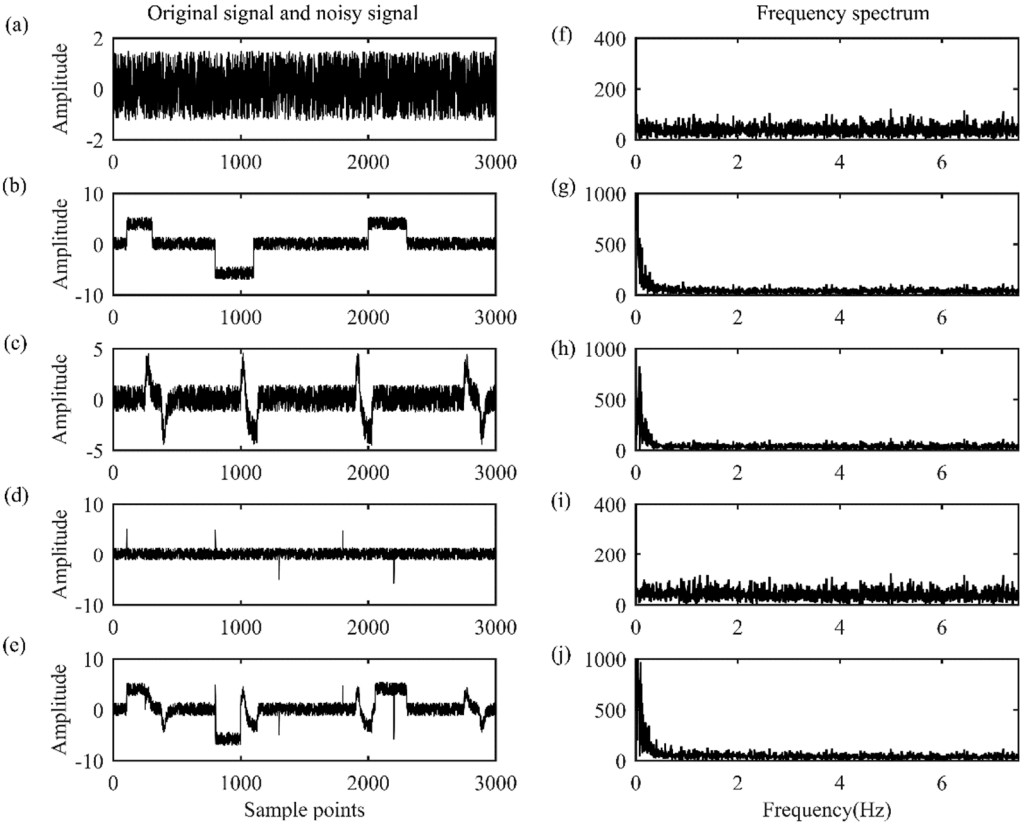

**Figure 2.** (**a**) Noise-free MT data, (**b**) data contaminated by square wave noise, (**c**) data contaminated by charge–discharge triangular wave noise, (**d**) data contaminated by impulse noise, (**e**) data contaminated by various noise. (**f–j**) is the frequency spectrum corresponding to (**a–e**). Noise-free MT data were collected in Linze Province with a sampling rate of 15 Hz. All the noise was randomly simulated.

*3.1. Entropy Analysis*

To reduce the loss of signal, multiscale dispersion entropy (*MDE*) was used to pre-process the overall data, with noisy data being selected. The *MDE* was calculated for the useful signal and noisy signal shown in Table 1 in order to test whether the difference between different signals could be found. This indicated that there was a sharp gap between the *MDE* of noise-free data and that of noisy data, meaning that *MDE* is capable of distinguishing different signals. Furthermore, the reason why *MDE* is more suitable as an evaluation criterion of noisy data is also shown in Table 1. The multiscale approximate entropy (MAE) and multiscale fuzzy entropy (MFE) of different signals are approximate (the difference is less than 0.05); thus, it was difficult to differentiate the noise from the useful signal. However, there was an obvious difference between the useful signal and noisy signal in *MDE* and multiscale sample entropy (MSE). Figure 3 shows their stability in different conditions. The green dashed line shows the baseline of the *MDE*, the blue dashed line shows the baseline of the MSE, and both of the values are shown in Table 1. The sample points below the baseline are considered to be noisy data. The red solid line shows the *MDE*, while the black solid line shows the MSE. The MSE in the useful signal fluctuates sharply (in Figure 3a). Compared with the data points filtered by the *MDE*, there are fewer data points filtered by MSE (In Figure 3b–e). Therefore, the *MDE* is more suitable to be used as an evaluation method for distinguishing between signals and noise. Meanwhile, various types of noise have a different performance of *MDE*. The signal with square wave noise has the smallest *MDE* value, meaning that it is the easiest to be selected using *MDE*; the triangle wave is second and the *MDE* of the signal with the addition of impulse noise is the closest to that of the useful signal.

**Table 1.** Entropy values of noise-free data and noisy data.

| Different Signals | MAE | MSE | MFE | MDE |
|---|---|---|---|---|
| Noise-free signal | 0.2531 | 1.6493 | 0.1582 | 2.7568 |
| With square wave | 0.4519 | 0.9279 | 0.1369 | 1.1521 |
| With triangle wave | 0.3458 | 1.3142 | 0.1174 | 2.0091 |
| With impulse noise | 0.3379 | 1.5034 | 0.1041 | 2.3361 |
| With various noise | 0.4323 | 0.9013 | 0.1124 | 1.1055 |

*3.2. Parameter Calculation*

After selecting noisy data, the parameters of the method should be calculated. In the proposed method, the wavelet decomposition level needs to be determined. The whole data set is divided into different data segments in order to calculate the *MDE* value for the evaluation of whether the data segment contains noise; thus, the number of noisy data segments is different for signals with the addition of different noise. There are six noisy data segments in the signal with square wave noise. There are 10 noisy data segments in other signals. The most common method used to determine the wavelet decomposition level is repeated experiments. In this method, the operator sets the value of the parameter one by one. To choose the best value, the four signal properties are calculated for each value. The result of the repeated tests is shown in Figure 4. When the number is 4 or 5, the improvement is obvious. Thus, the value is set as 4 or 5 by operators through repeated tests. To verify the reliability of the parameter calculated in the proposed method, the results determined by the proposed method in different data segments are shown in Table 2. These values are mainly around 4 or 5, and the same as those calculated by the common method. In other words, the values of the parameters calculated by the proposed method are accurate and this method is more efficient.

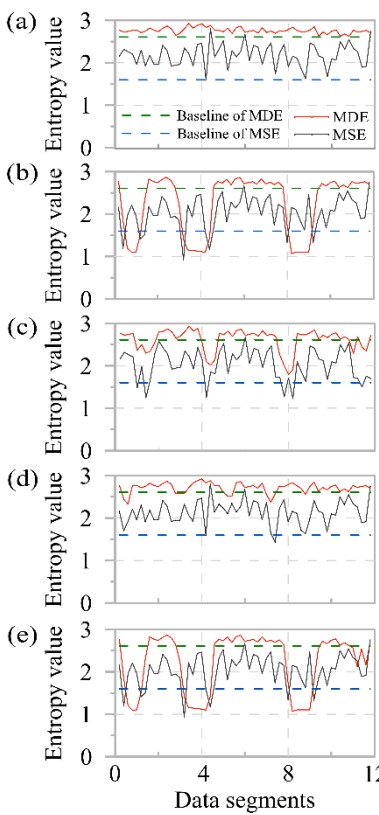

**Figure 3.** Entropy value of noise-free data and noisy data at different data segments. (**a**) Noise-free MT data, (**b**) data contaminated by square noise, (**c**) data contaminated by charge–discharge triangular wave, (**d**) data contaminated by impulse noise, (**e**) data contaminated by various noise. The green dashed line is the baseline of *MDE*, and the blue dashed line is the baseline of MSE. The red solid line shows the *MDE*, and the black solid line shows the MSE.

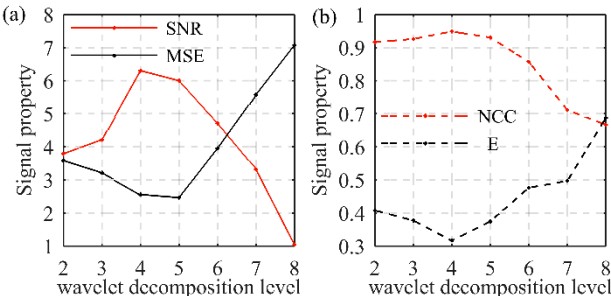

**Figure 4.** The choice of the best wavelet decomposition level made through repeated tests. (**a**) shows SNR and MSE of the whole noisy signal at different wavelet decomposition levels, (**b**) shows NCC and E of the whole noisy signal at different wavelet decomposition levels. Red solid line indicates SNR, black solid line indicates MSE, red dashed line indicates NCC, and black dashed line indicates E.

**Table 2.** The wavelet decomposition level calculated by the proposed method for the different noisy data segments in the different signals (signals with different types of noise contain different numbers of noisy data segments).

| Signals with Different Noise | Noisy Segment1 | Noisy Segment2 | Noisy Segment3 | Noisy Segment4 | Noisy Segment5 | Noisy Segment6 | Noisy Segment7 | Noisy Segment8 |
|---|---|---|---|---|---|---|---|---|
| Square wave | 4 | 5 | 4 | 5 | 4 | 5 | - | - |
| Triangle wave | 5 | 5 | 4 | 5 | 4 | 4 | 4 | 4 |
| Impulse noise | 4 | 5 | 4 | 4 | 4 | 5 | 4 | 4 |
| Various noise | 4 | 5 | 4 | 4 | 4 | 4 | 4 | 4 |

### 3.3. Performance Evaluation

In the section, we analyze the de-noising performance of the proposed method. Figure 5 shows that the results gained by extracting different types of noise using the proposed method. In Figure 5a–d, the signal is contaminated with different types of noise and the noisy data segments are completely selected by *MDE* with little loss of useful data around the noise. However, there is an inevitable loss of useful signal, which is overlapped in the noise, meaning that the noisy signal must be further processed by the proposed de-noising method. As shown in Figure 5e–h, the profiles of noise are extracted and the fluctuation of the profile of the noise is decreased after processing by the proposed method. Compared with processing all the data points, the proposed method with *MDE* can reduce the loss of useful signals. In Figure 5i–l, it is indicated that the frequency amplitude of the noise extracted by the proposed method is same as the amplitude of noisy data (see Figure 2g–j). Therefore, on the basis of reducing the useful signal, the proposed method is capable of removing both large-scale noise and impulse noise. However, compared with the triangular wave noise and impulse noise, the improvement of square wave noise achieved by the proposed method is better, with less loss of signals.

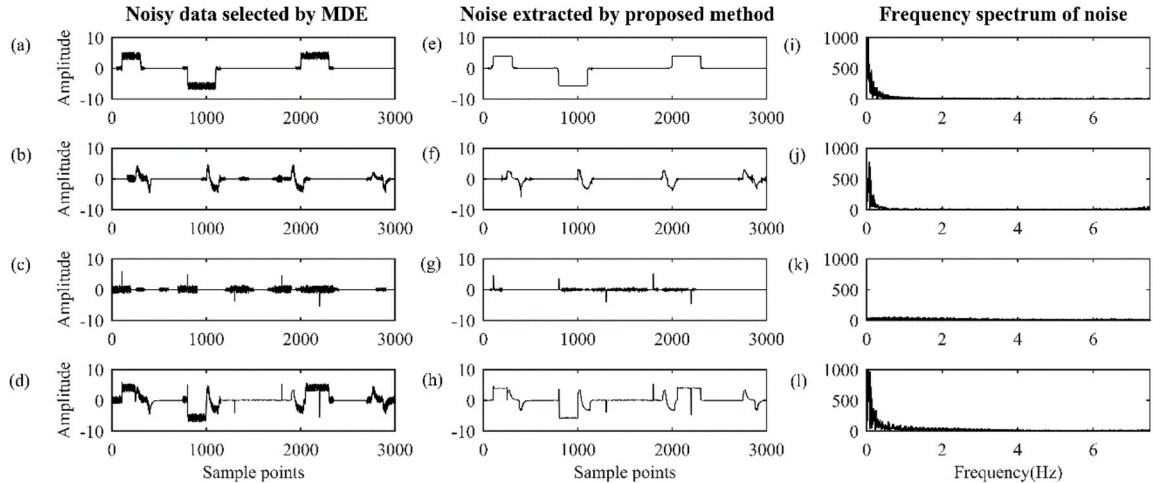

**Figure 5.** Data detected by *MDE* and the noise time–frequency spectrum extracted by the proposed method. (**a**) shows the noisy data selected by *MDE* in the signal contaminated by square wave noise, (**e**) indicates the square wave noise extracted by the proposed method. (**b**,**f**) are the results of the signal contaminated by charge–discharge triangular wave. (**c**,**g**) denote the results of data contaminated by impulse noise. (**d**,**h**) represent the results of data contaminated by various noise types. (**i**–**l**) are the frequency spectrum corresponding to (**e**–**h**).

Figure 6a–j show the time series after eliminating various types of noise using the proposed method; the corresponding frequency spectrum is shown in Figure 6k–o. Due to the lack of noisy data, the result of Figure 6f is the same as that of the original signal (see Figure 6a). The time series after eliminating the square wave noise is shown in Figure 6g, where it can be seen that the profile of the useful signal is similar to that shown in Figure 6a. The results after suppressing the triangular wave noise and various other types of noise are shown in Figure 6h,j. It is evident that there are losses around 500 data point and 2900 data point (see red boxes and blue boxes), with fewer losses seen in Figure 6j. In other words, when there are various different types of noise in the MT data simultaneously, the improvement gained by removing triangular waves through the proposed method is better. Figure 6i shows the time series after eliminating the impulse noise, which is roughly the same as the noise-free data.

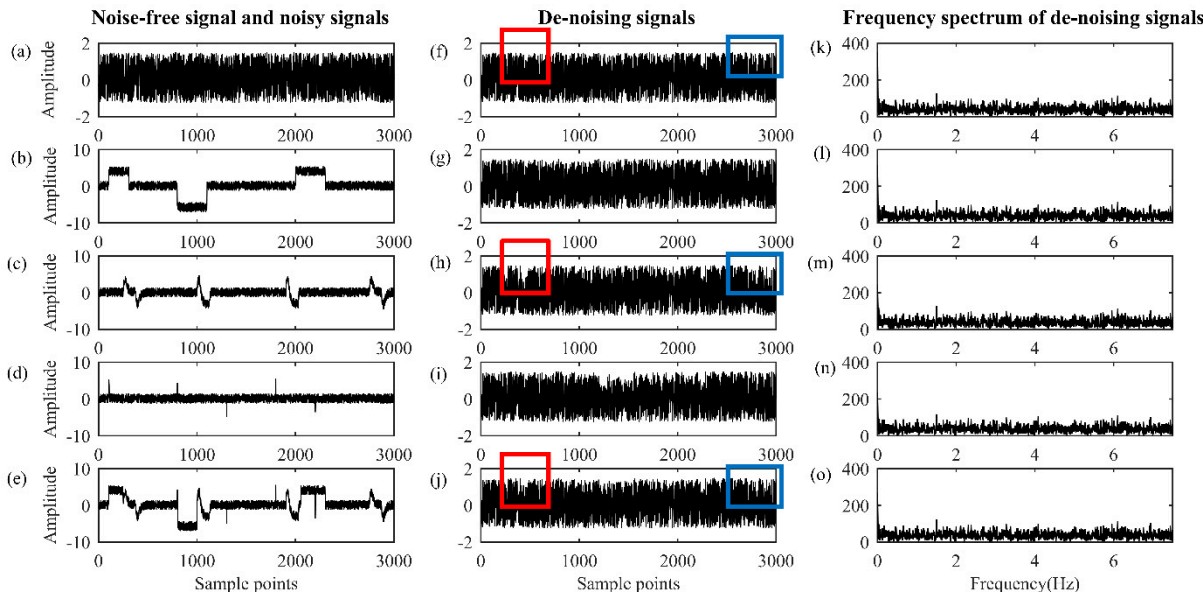

**Figure 6.** Time–frequency domain results after application of the proposed method. (**a**–**j**) shows the original and de-noised time series of noise-free MT data, data contaminated by square noise, data contaminated by charge–discharge triangular waves, data contaminated by impulse noise, data contaminated by various types of noise. (**k**–**o**) is the frequency spectrum corresponding to (**f**–**j**). Red boxes indicate the differences between the de-nosing signal and original signal at 500 data point. Blue boxes indicate the differences between the de-nosing signal and original signal at 2900 data point.

The improvement results after de-noising through singular value decomposition (SVD), wavelet transform (WT), and the proposed method (S-W) are shown in Figure 7. Compared with the results gained using the above two methods for different types of noise, it can be seen that the proposed method removes the profiles of noise more completely. For large-scale noise, SVD decomposition and wavelet transform find it difficult to suppress the interference, with residual noise being left (see blue lines and red lines in Figure 7a–c,j–l). For impulse noise, the proposed method also achieved better results than the other methods (In Figure 7d,e). However, compared with the time series seen after eliminating various types of noise in Figure 7j–l, it can be seen that only subtle differences are achieved from eliminating one type of noise through different methods. Thus, we introduced the use of a frequency spectrum to show the differences between the methods (see Figure 8). Figure 8d–f show the frequency spectrum after impulse noise was removed. The amplitude is similar because impulse noise has little influence on the frequency domain. For large-scale noise, obvious differences can be seen in the 0–1 Hz frequency band (see green lines in Figure 8a–c,g–i,j–l). The amplitude of the spectrum after SVD decomposition and wavelet transform were performed is greater than that after the proposed method was performed. In other words, it is difficult for the two above methods to completely suppress various types of noise.

As shown in Figure 9, four signal properties were calculated to evaluate the de-noising performance. The green line represents the proposed method, the red line shows the SVD decomposition, and the black line indicates the wavelet transform. With the increase in S/N (namely, the level of noise decreased), both the SNR and NCC increased correspondingly and the E and MSE gradually decreased. The values of the signal properties in the proposed method were always better than those of wavelet transform and SVD decomposition. Therefore, the proposed method has more advantages for improving the signal-to-noise ratio.

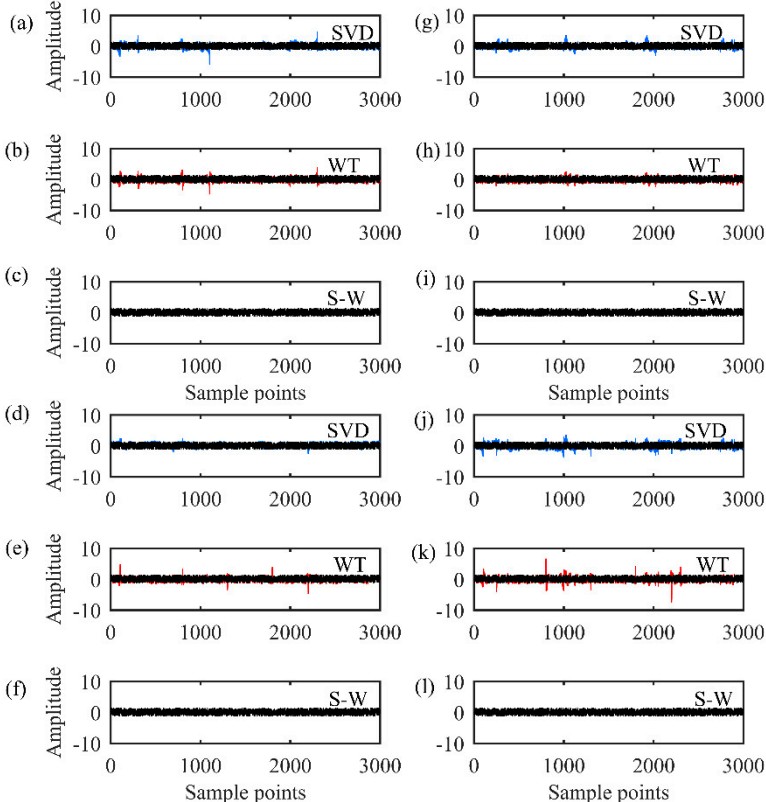

**Figure 7.** The de-noising results obtained after the use of different methods. Black lines represent the original signal as a reference, blue lines indicate the de-noising results of SVD, red lines denote the processing results of WT, purple lines show the results improved by the proposed method, completely covered by black lines. (**a**–**c**) show the time series of data contaminated by square noise after the use of SVD, WT, and the proposed method. (**d**–**f**) show the time series of data contaminated by impulse noise after the above different methods have been used. (**g**–**i**) show the time series of data contaminated by charge–discharge triangular waves after the above different methods have been used. (**j**–**l**) shows the time series of data contaminated by various types of noise after the above different methods have been used.

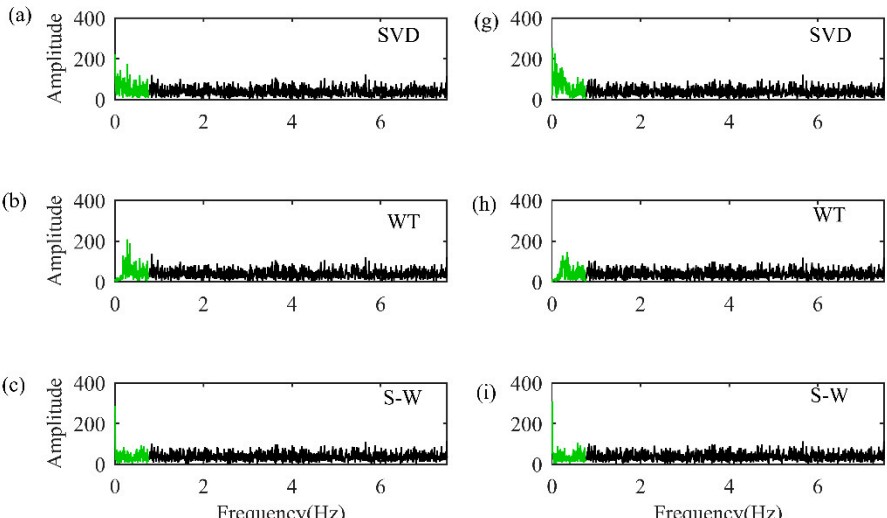

**Figure 8.** *Cont.*

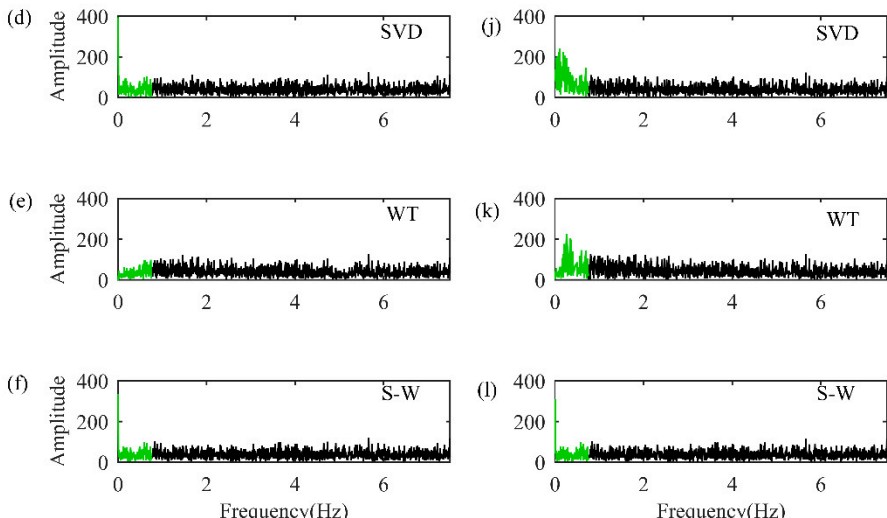

**Figure 8.** Frequency domain results after the use of the different methods. (**a–c**) show the spectrum of data contaminated by square noise after the use of SVD, WF, and the proposed method. (**d–f**) show the spectrum of data contaminated by impulse noise after the above different methods have been used. (**g–i**) show the spectrum of data contaminated by charge–discharge triangular waves after the above different methods have been used. (**j–l**) show the spectrum of data contaminated by various types of noise after the above different methods have been used.

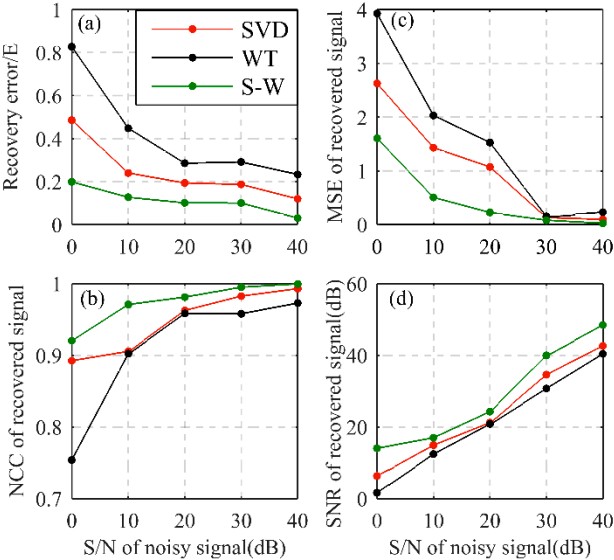

**Figure 9.** (**a**) Recovery error (E), (**b**) NCC, (**c**) MSE, and (**d**) SNR of the signals recovered by different methods at different S/Ns. Black line indicates wavelet transform, red line denotes SVD decomposition, and green line shows the proposed method.

## 4. Implementation for MT Field Data

In this section, the MT filed data are applied to verify the practicability of the proposed method to improve the MT response, which is measured in Qilian County, Qinghai Province, with a sampling rate of 15 Hz and a distance between the two sites of 10 km. The survey area is quite far away from the urban area; thus, the data quality of most of the sites is good. However, site 380 is close to the industrialized area and the data are contaminated by various noises, resulting in a low signal-to-noise ratio.

We randomly selected from the sites with high-quality data, and then took site 300 and site 380 as examples. As shown in Figure 10, the results obtained after removing different noise levels are compared with the raw time series. In Figure 10a, it can be seen that the

field data of site 380 are contaminated by square wave noise and triangular wave noise. Figure 10b shows the profile extracted by the proposed method, which indicates that the noise is completely abstracted and that the trend of the useful signal is recovered after processing using the proposed method (see Figure 10a,c). In Figure 10d, the signal-to-noise ratio of site 300 is high; the data are only contaminated by impulse noise, which can also be eliminated using the proposed method. Thus, the improvement obtained by the proposed method is effective, whether the data are measured with a high noise level or a low noise level.

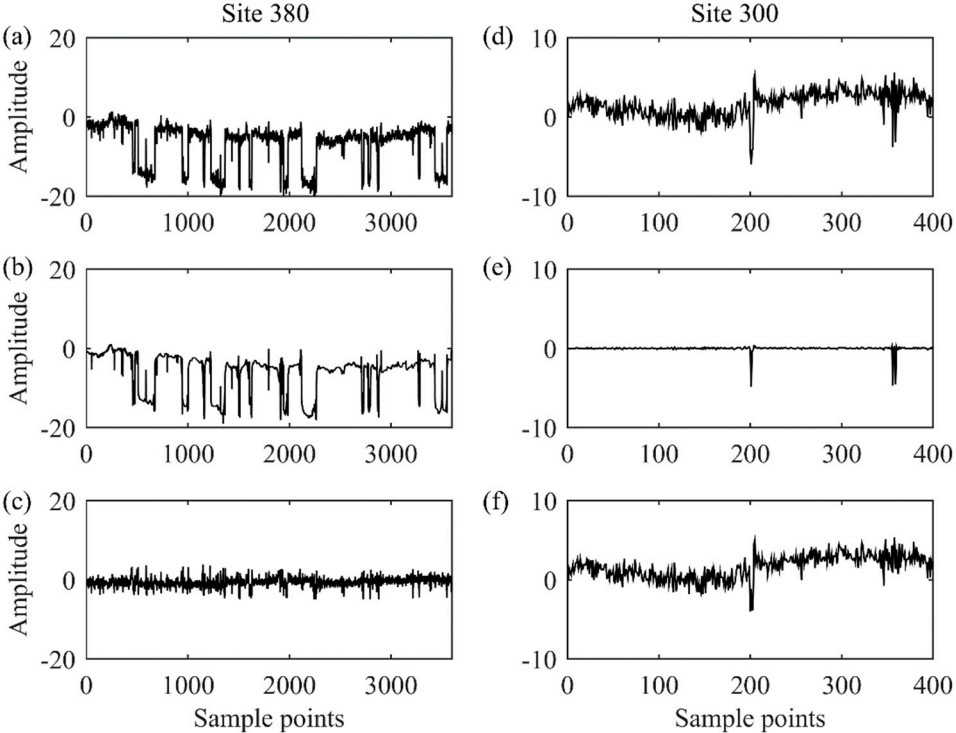

**Figure 10.** Time series segments of the real sites 380 and 360 in Qilian area with a sampling rate of 15 Hz. (**a**) Raw signal of Site 380, (**b**) noise extracted by the proposed method, and (**c**) signal de-noised by the proposed method. (**d**) Raw signal of site 360, (**e**) noise extracted by the proposed method, and (**f**) signal de-noised by the proposed method.

Figure 11 shows the improvement of site 380 through SVD decomposition, wavelet transform, and the proposed method. However, when large-scale noise is subtracted in the previous two methods, there is still impulse noise remaining (see Figure 11b–d). Meanwhile, in the 0–2 Hz (Figure 10f–h) region, it can be seen that the amplitude of the SVD decomposition and wavelet transform is more than that of the proposed method, showing that the signal and noise separation in the proposed method is better and more effective.

The apparent resistivity phase curves of site 380, site 360, site 340, site 320, site 300, and site 280 processed using the robust method and the proposed methods are shown in Figure 12. There is are high-quality data for site 360, site 320, and site 300, which is less contaminated by noise. For these sites, the results of the robust method and the proposed method are almost the same, which indicates that the improvements made by the proposed method are reliable and reasonable. However, when the measured data are contaminated with a higher noise level (which is the case for site 340, site 280, and site 380), the robust method can only improve the fluctuation of some jump points, indicating that the improvement is weaker than that gained in the proposed method. For poor-quality data, the proposed method is able to recover the smoothness of the apparent resistivity phase curves. Although the improvement of the proposed method is more obvious for $\rho_{yx}$ in site 340 and $\rho_{xy}$ in site 280, the de-noising curves of two methods for $\rho_{xy}$ in site 340 and

$\rho_{yx}$ in site 280 are overlapping, which further verifies the stability and effectiveness of the proposed method.

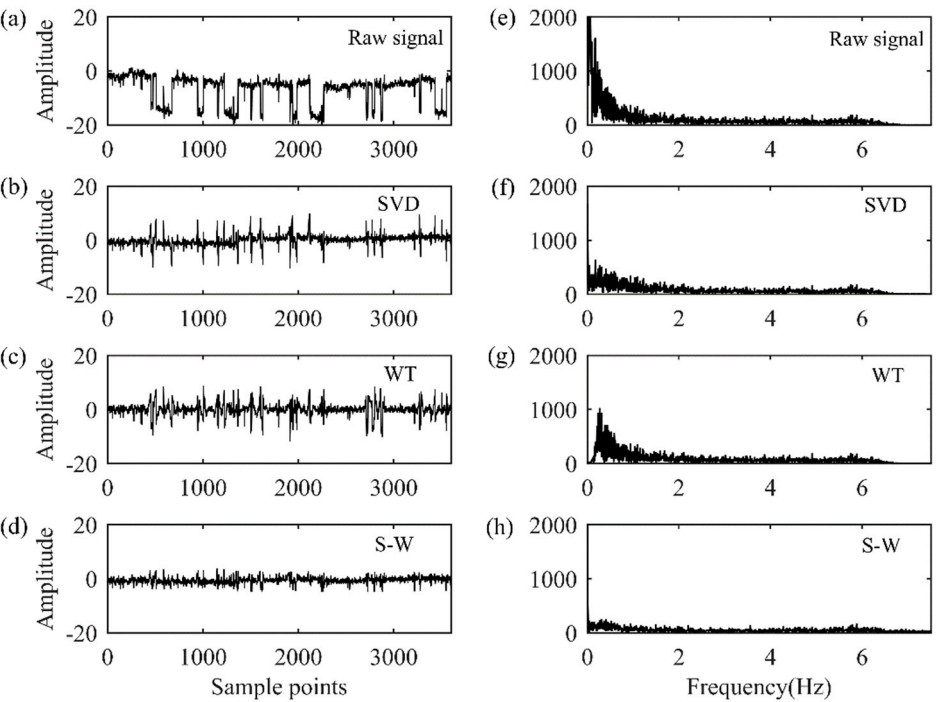

**Figure 11.** Time series segment of site 380 after the use of different methods. (**a**) Raw signal of site 380, (**b**) noise de-noised by SVD, (**c**) signal de-noised by WT, (**d**) noise de-noised by the proposed method, and (**e**–**h**) the frequency spectrum corresponding to (**a**–**d**).

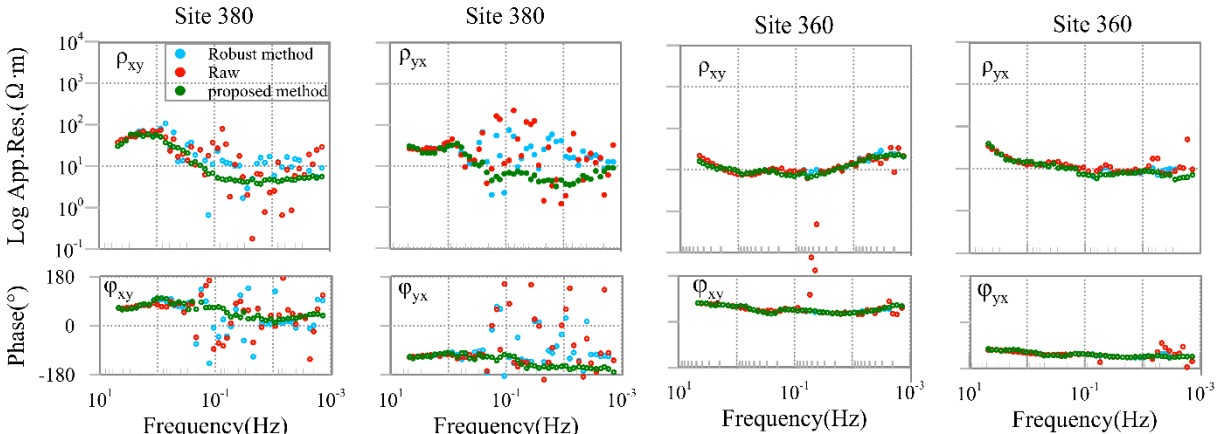

**Figure 12.** *Cont.*

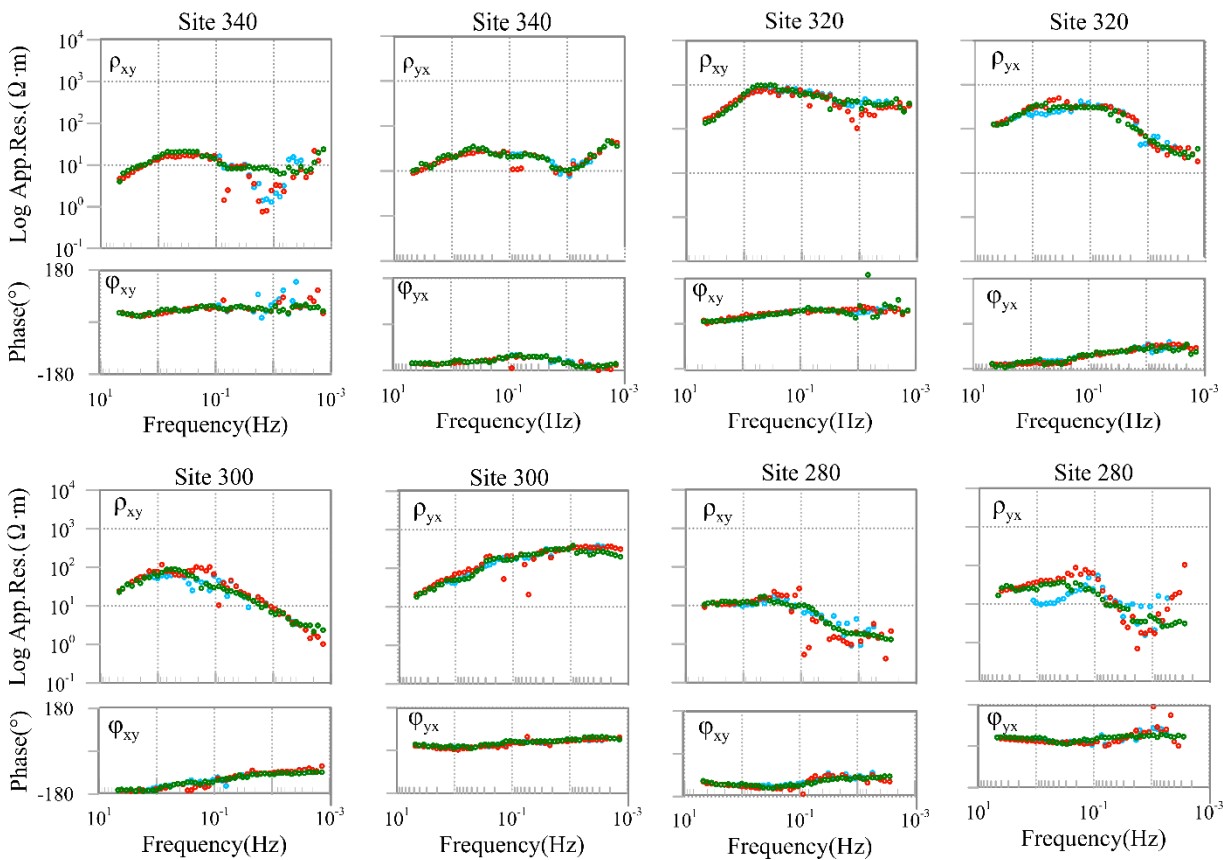

**Figure 12.** Apparent resistivity phase curves of field sites. The red curves represent the results obtained using raw data. The green curves denote the results obtained using the data de-noised using the proposed method. The blue curves stand for the results obtained using the data de-noised by the robust method.

## 5. Conclusions

To solve the MT response distortions caused by high levels of noise, we propose the use of a de-noising method based on discrete wavelet transform and singular value decomposition (SVD) that consists of three sections. Firstly, multiscale dispersion entropy (*MDE*) selects noisy data to reduce the loss of useful signal. Then, phase space reconstruction calculates the values of parameters in the method. Finally, discrete wavelet transform and iterative SVD decomposition realize the signal–noise separation in MT data.

Compared with the traditional SVD decomposition and wavelet transform achieved in synthetic tests, the proposed method is more able to eliminate various types of noise without noise remaining. Meanwhile, the data measured in the Qilian area were used to test the reasonability of the proposed method. Our method improves the quality of the data measured in the Qilian area. Furthermore, based on the more continuous and smoother response curves gained using the proposed method than the robust estimation, we can deduce that the proposed method is more effective and more suitable for improving the quality of data with a low signal-to-noise ratio.

The proposed method has many advantages for processing poor-quality data; thus, it can be applied for the de-noising of MT data in environments with strong interference (such as industrialized areas). In comparison with conventional methods, the parameters of the proposed method are selected adaptively according to the characteristics of the data with no need for manual intervention, which reduces the amount of time needed.

**Author Contributions:** Conceptualization, R.Z. and J.H.; methodology, R.Z., J.H. and Z.G.; software, R.Z.; formal analysis, R.Z., J.H. and Z.G.; investigation, R.Z.; writing—original draft preparation, R.Z.; writing—review and editing, R.Z., J.H., T.L. and Z.G.; visualization, R.Z.; funding acquisition, J.H. All authors have read and agreed to the published version of the manuscript.

**Funding:** This paper is financially supported by the National Key Research and Development project(2017YFC0601305), the Second Tibetan Plateau Scientific Expedition and Research (STEP) program (Grant No. 2019QZKK0701), the National Natural Science Foundation of China (41504076) and the Jilin Province Geological Exploration Fund Project (2018-19).

**Conflicts of Interest:** The authors declare no conflict of interest.

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
