# Peer review of "De-Noising of Magnetotelluric Signals by Discrete Wavelet Transform and SVD Decomposition"

_remotesensing, doi:10.3390/rs13234932_

Round 1

Reviewer 1 Report

The manuscript “De-noising of magnetotelluric signals by discrete wavelet transform and SVD decomposition” is dealing with a  data processing technique based on wavelet filtering and SVD decomposition applied to magnetotelluric data.

Despite the topic might be interesting, the paper is poorly prepared: 
not only the English level makes it hard to follow the Authors’ points (it is not only a matter of grammatical errors, several sentences, even if with correct syntaxes, are not really meaningful), but the theoretical framework is barely discussed and often symbols are not properly introduced.
On the same line of reasoning, also the figures are not effectively discussed in the text and also captions are not always easy to be interpreted.

So, despite the potential relevance of the research, I highly suggest the Authors prepare a deeply revised version of their manuscript. 

In the attached pdf, more detailed comments from my side can be found.
I hope this can help to improve the quality of the paper.

Author Response

Response to Reviewer 1 Comments

Thank you for your affirmation of our manuscript, which gives us the motivation to research and write. This time we have made some revision to the paper, hoping to get your approval.

The revised parts in the manuscript were indicated in red (please see the attachment).

Point 1: Introduction poor English level. Sentences might be correct from a grammatical perspective, but, often, they are very vague and not really meaningful.

Response 1: We have submitted the manuscript to MDPI for the translation, and rewritten the vague sentences in the introduction. The revised parts in the introduction were indicated in red.

Point 2: Advise to change the word ‘destroy’ to ‘distortions’.

Response 2: The word of ‘distortions’ is more suitable for the grammar of the sentence. We have modified this word. The revised part in the introduction was indicated in red.

Point 3: ‘Obviously, the methods above have limitations such as the levels of noise in MT data, which are ineffective when signal-to-noise ratio is low.’ What does this mean?

Response 3: Through this sentence, we wanted to indicate the disadvantages of the existing de-noising methods: When the noise level is high, the improvement is poor. We have modified this sentence. The revised parts in the introduction were indicated in red.

Point 4: In section 2.1, no real explanations of the formula. Possibly several typos in the equations.

Response 4: We appreciate the suggestion and have modified the question in section 2.1. This suggestion is very helpful to us. We have added the relevant explanations of the equations. The revised parts in section 2.1 were indicated in red, including eq.1, eq.2, eq.4 and eq.5.

Point 5: In eq.2, Where is the dependency from t? what is x?

Response 5: We are very sorry for incorrectly writing the formula, which has been modified in the eq.2. t is the length of observation time in each data segment and x is the data points in each data segment.

Point 6: ‘For the selection of m, which must satisfy m>2d+1 in Takens theory, where d represents the real dimension of attractors.’ No meaning. To be rephrased.

Response 6: In this sentence, we want to denote that the value of m should meet the formula of Takens theory (that is m>2d+1), which is a necessary condition. We have modified this sentence. The revised parts in the section 2.2 were indicated in red.

Point 7: It is unclear why the type of used wavelet shouldn't matter.

Response 7: We appreciate the professional suggestion of the reviewer. There is something unclear in our words. The selection of mother wavelet function is important, but scholars had done the relevant research on its selection, and come to the conclusion that coif wavelet function is more suitable for MT signal processing. Therefore, we quote the previous research results to solve the selection of the mother wavelet function. We have modified this sentence. The revised parts in the section 2.3 were indicated in red.

Point 8: In Fig.3, the associated caption is incomplete. It is not clear what is what.

Response 8: We have modified this question. All the information in Fig.3 is explained in its caption, and we have modified Fig.3 to make readers more easily understand the meaning of Fig.3. The revised parts in the Fig.3 were indicated in red.

Point 9: ‘In Table 1, it shows average entropy of the useful signal and noisy signals, to obtain the differences between various entropy values in the signals.’ to be rephrased.

Response 9: We have modified this question to express this sentence more clearly and correctly. The revised parts in the section 3.1 were indicated in red.

Point 10: In Fig.5, what are the top rows?

Response 10: We appreciate the reviewer for reminding us of what we might neglect in the original manuscript. In Fig.5, the top rows are the processing results of noise-free data. We originally wanted to use the top rows to show that our method would not lose the useful signal, but the meaning was not clearly expressed, so the top rows are meaningless. We have modified this question with deleting the top rows. The revised parts in the Fig.5 were indicated in red.

Special thanks to you for your recognition.

We appreciate for Editors/Reviewers’ warm work earnestly, and hope that the correction will meet with approval.

Once again, thank you very much for your comments and suggestions.

Reviewer 2 Report

Our paper is very interesting, however, I believed that you may have not submitted it to the best journal. 

The aims of the Remote Sensing Journal is "Remote Sensing (ISSN 2072-4292) publishes regular research papers, reviews, technical notes and communications covering all aspects of remote sensing science, from sensor design, validation/calibration, to its application in geosciences, environmental sciences, ecology and civil engineering". 

Author Response

Response to Reviewer 2 Comments

Thank you for your affirmation of our manuscript, which gives us the motivation to research and write. This time we have made some revision to the paper, hoping to get your approval.

The revised parts in the manuscript were indicated in red (please see the attachment).

Special thanks to you for your recognition.

We appreciate for Editors/Reviewers’ warm work earnestly, and hope that the correction will meet with approval.

Once again, thank you very much for your comments and suggestions.

Reviewer 3 Report

First, I would like to congratulate the authors on the idea of interesting analyzes. The subject is interesting and worth researching. The problem of measuring compatibility and filtering out their distortions is very important. At the beginning, the authors described the current state of knowledge, characterizing individual numerical methods. Then they analyzed selected solutions. The conclusions from the analyzes were correctly drawn. In my opinion, the article is suitable for publication in this journal.

Author Response

Response to Reviewer 3 Comments

Thank you for your affirmation of our manuscript, which gives us the motivation to research and write. This time we have made some revision to the paper, hoping to get your approval.

The revised parts in the manuscript were indicated in red (please see the attachment).

Special thanks to you for your recognition.

We appreciate for Editors/Reviewers’ warm work earnestly, and hope that the correction will meet with approval.

Once again, thank you very much for your comments and suggestions.

Reviewer 4 Report

The paper proposes a method for noise detection in magneto telluric data based on discrete wavelet transform and singular value decomposition. The authors claim that the proposed de-noising method allows determining the number of wavelet decomposition layers, while suppressing noise more effectively.

I suggest a to made some changes based on the following points:

1.- English grammar and style must be revised.

2.- Please describe the meaning of SVD the first time it appears in the text.

3.- Introduction section. The authors must clearly state the novelties and contributions of this work and the progress achieved with respect to similar works. If necessary, add the text as bullet points.

4.- Section 2.1. The following paragraph is not clear, please rewrite: “Compared with multiscale sample entropy (MSE), multiscale fuzzy entropy (MFE), multiscale approximate entropy (MAE), the calculation of MDE is simpler and faster”

5.- The modulus of eq. (20,21,23,24) has a subscript 2. Please explain the meaning and also describe the meaning of (24) S/N

6.- Figure 2 (b) to (e) correspond to different noise types. Have they been artificially generated? In case of affirmative answer, which is the interest? Are these noise types and levels representative of real situations? Why do not directly use real noisy signals?

7.- There are few references from 2020-2022

8.- My main concern is the practical applicability of the results and methods presented in this paper. This is a critical point. The authors must do an effort in developing this point.

I believe the remarks above would help to improve the paper.

Author Response

Response to Reviewer 4 Comments

Thank you for your valuable comments on our article. We have made corresponding amendments to your questions and hope to get your satisfactory reply. The following are detailed responses:

The revised parts in the manuscript were indicated in red (please see the attachment).

Point 1: English grammar and style must be revised.

Response 1: We have made the overall modification to the grammar and style of the manuscript and submitted it to MDPI for polishing.

Point 2: Please describe the meaning of SVD the first time it appears in the text.

Response 2: We have modified this question when SVD appears in the abstract, the introduction and the conclusion for the first time. The revised parts in the abstract, the introduction and the conclusion were indicated in red.

Point 3: Introduction section. The authors must clearly state the novelties and contributions of this work and the progress achieved with respect to similar works. If necessary, add the text as bullet points.

Response 3: We appreciate the reviewer for reminding us of what we might neglect in the original introduction. We have modified this question. We have added the texts to highlight the advantages and contribution of our method in the introduction. The revised parts in the introduction were indicated in red.

Point 4: Section 2.1. The following paragraph is not clear, please rewrite: “Compared with multiscale sample entropy (MSE), multiscale fuzzy entropy (MFE), multiscale approximate entropy (MAE), the calculation of MDE is simpler and faster”.

Response 4: We have modified this question. The revised parts in the section 2.1 were indicated in red.

Point 5: The modulus of eq. (20,21,23,24) has a subscript 2. Please explain the meaning and also describe the meaning of (24) S/N.

Response 5: We have modified this question. ||•||2 is 2-norm function, S/N denotes the level of the added noise, and we have explained the meaning of different values of S/N in detail. The revised parts in the section 3 were indicated in red.

Point 6: Figure 2 (b) to (e) correspond to different noise types. Have they been artificially generated? In case of affirmative answer, which is the interest? Are these noise types and levels representative of real situations? Why do not directly use real noisy signals?

Response 6: We appreciate the professional suggestions of the reviewer. In the synthetic tests, different noise types are all artificially generated. They can reflect the real situations because they all can be find in MT time-series (as shown in the following Figure 1–2). In Figure 1, red boxes show the noise consisting of square wave noise, charge–discharge triangular wave noise and impulse noise. Green boxes show the noise consisting of square wave noise and impulse noise. Blue boxes show the noise consisting of charge–discharge triangular wave noise and impulse noise. Various noise types are overlapped in Figure 1. In Figure 2, yellow boxes show the square wave noise, purple boxes show the charge–discharge triangular wave noise and pink boxes show the impulse noise. Different noise types are closely adjacent in Figure 2. The reasons why we are not use real noisy signals and the advantages of using artificial noise are as follows:

In general, the real noise exists in complex situations, consisting of various noise types (see Figure 1–2), and it has always been no solution to control the intensity of the real noise. It is difficult to obtain the single noise type. In the synthetic tests, we want to use a single type of noise to evaluate the identification and de-noising performance of our method for each noise type, which cannot be realized by real noise. The last type of noise in the manuscript is the mixed noise, which is corresponding to the actual situation in section 4.

There is something wrong for us to upload the Figure1-2 here, thus, the Figure 1-2 are put in the appendix of the revised manuscript in the attachment.

Figure 1. Various noise types in the MT field data of site 380 in Qilian area. Red boxes show the noise consisting of square wave noise, charge–discharge triangular wave noise and impulse noise. Green boxes show the noise consisting of square wave noise and impulse noise. Blue boxes show the noise consisting of charge–discharge triangular wave noise and impulse noise.

There is something wrong for us to upload the Figure1-2 here, thus, the Figure 1-2 are put in the appendix of the revised manuscript in the attachment.

Figure 2. Different noise types in the MT field data of site 128 in Shuangjianzishan area. Yellow boxes show the square wave noise. Purple boxes show charge–discharge triangular wave noise. Pink boxes show the impulse noise.

Point 7: There are few references from 2020-2022.

Response 7: We have adjusted the references and added relevant references from 2020 to 2022. The revised parts in the reference were indicated in red.

Point 8: My main concern is the practical applicability of the results and methods presented in this paper. This is a critical point. The authors must do an effort in developing this point.

Response 8: We have studied the reviewer’s comment carefully. As the reviewer mentioned, the practicability of our method is a very important point. We have applied our method in the MT field data of Linze area in synthetic tests and the MT field data of Qilian area in section 4. All of the results processed by our method are better than the conventional methods. Meanwhile, we have compared the results of our method with those of the robust method, which is considered to be reliable in MT de-noising methods. The improvements of the two methods are the same for high-quality data. For the data with poor-quality in Qilian area, our method makes the response curves more continuous and smoother, which is meaningful to image the subsurface electromagnetic structure. The revised parts in the manuscript were indicated in red, including section 3 and section 4.

Special thanks to you for your good comments.

We appreciate for Editors/Reviewers’ warm work earnestly, and hope that the correction will meet with approval.

Once again, thank you very much for your comments and suggestions.

Reviewer 5 Report

Dear Authors,

I think it is of an author's best interest to have a review with the highest amount of fair-criticism as possible, thus having his/her name associated with high-quality work. Minding the time constraints to review this paper, I spent the maximum amount of time I could on it and tried to be as critical as I could.

As a Reviewer my general opinion is that the topic is interesting.

Your work is well-written and well-presented.

1) Why you use bold in your equations? Please fix it.

2)In Page 9, please fix the way you present the proposed method. It will upgrade your work.

3)Please explain more the metrics and numbers in Tables 1 and 2.

4)Please fix your Figures. Add more color in order to be more understandable.

5) Your conclusion is not well-written. Please re-write it in order to help the reader understand the aim of your work.

Author Response

Response to Reviewer 5 Comments

Thank you for your comments concerning our manuscript. Those comments are all valuable and very helpful for revising and improving our paper, as well as the important guiding significance to our researches. Hoping to get your approval. The main corrections in the paper and the responds to the reviewer’s comments are as following:

The revised parts in the manuscript were indicated in red (please see the attachment).

Point 1: Why you use bold in your equations? Please fix it.

Response 1: We have modified these equations. The revised equations in the manuscript were indicated in red, including eq.16, eq.18 and eq.19.

Point 2: In page 9, please fix the way you present the proposed method. It will upgrade your work.

Response 2: We appreciate the professional suggestions of the reviewer. We have adjusted the order of sentences, deleted unnecessary sentences, added the texts about the interpretation, and modified the pictures, so that the way of proposing our method becomes more logical and clearer. Because we have modified the format of the manuscript, page 9 in the original manuscript has changed to page 11 in the revised manuscript. The revised parts in the page 11 were indicated in red.

Point 3: Please explain more the metrics and numbers in Tables 1 and 2.

Response 3: We have modified this question, explaining more the metrics and numbers in Tables 1 and 2.

Table 1 shows entropy values can distinguish noise and the useful signal. The top row in Table 1 is names of different entropy types. The first column indicates the signals contaminated by different noise types. According to the corresponding entropy name, the numbers denote the entropy value of each signal.

Table 2 indicates the wavelet decomposition level calculated by the proposed method for different noisy data segments in the different signals. The top row in Table 2 is the various noisy segments in different signals. The first column shows the signals contaminated by different noise types. In different signals, the numbers denote the wavelet decomposition level calculated by the proposed method for each noisy segment.

Point 4: Please fix your Figures. Add more color in order to be more understandable.

Response 4: We appreciate the reviewer for reminding us of what we might neglect in the original manuscript. We have modified this question. We have used different colors to indicate the de-noising results of different methods. The differences of different methods are highlighted by the color. And we have added symbols to make Figures more understandable. The revised parts in the manuscript were indicated in red, including Figure 2, Figure 3, Figure 5, Figure 6, Figure 7 and Figure 8.

Point 5: Your conclusion is not well-written. Please re-write it in order to help the reader understand the aim of your work.

Response 5: We appreciate the suggestion and have modified the question in the conclusion. This suggestion is very helpful to us. The revised parts in the conclusion were indicated in red.

Special thanks to you for your good comments.

We appreciate for Editors/Reviewers’ warm work earnestly, and hope that the correction will meet with approval.

Once again, thank you very much for your comments and suggestions.

Round 2

Reviewer 1 Report

The Authors put limited effort into the preparation of a new version of the paper and definitely have not proceeded with an in-depth revision of their manuscript.

For these reasons, I cannot do anything else than confirm my previous suggestion: the manuscript should be largely revised and rearranged since, as it is now, I can hardly see the overall point of it

Author Response

Response to Reviewer 1 Comments

Review Report (Round 2)

Thank you for your comments concerning our manuscript. Those comments are all valuable and very helpful for revising and improving our paper, as well as the important guiding significance to our researches. Hoping to get your approval. The main corrections in the paper and the responds to the reviewer’s comments are as following:

The revised parts in the manuscript were indicated in red.

Point 1: The manuscript should be largely revised and rearranged

Response 1: We appreciate the professional suggestions of the reviewer. We have made the overall modification to the manuscript. The revised parts were indicated in red. The modification can be divided into six parts.

1) In the abstract, for the logical framework of this manuscript, we have revised the contents of the abstract. In the abstract, we outlined the status and existing problems of MT de-noising methods. Then, we summarized the results of synthetic tests, as well as the advantages of our method when comparing with other methods in the results of MT field data.

2) In the introduction, we rearranged the theoretical framework. Firstly, we introduced the theory of MT and its problems. Then, for these problems, various de-noising methods were proposed by scholars. However, there were some shortages in the above methods. Subsequently, for overcoming the above shortages, we proposed our method. Finally, we introduced our method briefly. Meanwhile, in the end of introduction, we have added the texts to highlight the advantages and contribution of our method.

3) In the section of method, we have modified some formulas and added necessary explanations of parameters. Meanwhile, we deleted meaningless sentences to make it easier for readers to understand the theory of our method.

4) In the synthetic tests, we have rephrased the sentences as well as modified the pictures and the description of the pictures, to make this section more logical and enable readers to more intuitively see the advantages of our method. Take three changes as examples: (1) In the page 11, we have adjusted the order of sentences, rephrased the necessary sentences, added the texts about the interpretation, and modified the pictures, so that the way of proposing our method becomes more logical and clearer. (2) We explained more the metrics and numbers in Tables 1 and 2. Table 1 shows entropy values can distinguish noise and the useful signal. The top row in Table 1 is names of different entropy types. The first column indicates the signals contaminated by different noise types. According to the corresponding entropy name, the numbers denote the entropy value of each signal. Table 2 indicates the wavelet decomposition level calculated by the proposed method for different noisy data segments in the different signals. The top row in Table 2 is the various noisy segments in different signals. The first column shows the signals contaminated by different noise types. In different signals, the numbers denote the wavelet decomposition level calculated by the proposed method for each noisy segment. (3) We have used different colors to indicate the de-noising results of different methods. The differences of different methods are highlighted by the color. And we have added symbols to make the figures more understandable. The other revised parts in the section 3 were also indicated in red.

5) In the section of implementation for MT field data, we have rephrased the sentences about the results of our method. All of the results processed by our method are better than the conventional methods. Meanwhile, we have compared the results of our method with those of the robust method which is considered to be reliable in MT de-noising methods. The improvements of the two methods (the robust method and our method) are the same for high-quality data. For the data with poor-quality in the Qilian area, our method makes the response curves more continuous and smoother, which is meaningful to image the subsurface electromagnetic structure. The revised parts in section 4 were indicated in red.

6) In the conclusion, for the logical framework of this manuscript, we have revised the contents of the conclusion. In the conclusion, firstly, we described the steps of our method. Then, through an overview of the work in the manuscript, it denotes our method is more effective for noise with high level, which shows that our method can be widely used in processing data measured in industrialized areas.

Review Report (Round 1)

Thank you for your valuable comments on our article. We have added texts about questions of round 1 in detail and hope to get your satisfactory reply. The following are detailed responses:

Point 1: Introduction poor English level. Sentences might be correct from a grammatical perspective, but, often, they are very vague and not really meaningful.

Response 1: We have submitted the manuscript to MDPI for the translation, and rewritten the vague sentences in the introduction. The revised parts in the introduction were indicated in red.

Point 2: Advise to change the word ‘destroy’ to ‘distortions’.

Response 2: The word of ‘distortions’ is more suitable for the grammar of the sentence. We have modified this word and rephrased the sentence. The revised parts in the introduction were indicated in red. The revised parts in the introduction were as follows:

‘However, during the period of observation, MT data are vulnerable to distortions from electromagnetic wave emitted by high-voltage lines, communication radio stations, and underground mining machines around monitoring sites.’

Point 3: ‘Obviously, the methods above have limitations such as the levels of noise in MT data, which are ineffective when signal-to-noise ratio is low.’ What does this mean?

Response 3: Through this sentence, we wanted to indicate the disadvantages of the existing de-noising methods: When the noise level is high, the improvement of the existing de-noising methods is poor. We have modified this sentence. The revised parts in the introduction were indicated in red. The revised parts in the introduction were as follows:

‘Obviously, these methods are ineffective in improving the quality of data when the signal-to-noise ratio is low’

Point 4: In section 2.1, no real explanations of the formula. Possibly several typos in the equations.

Response 4: We appreciate the suggestion and have modified the question in section 2.1. This suggestion is very helpful to us. We have added the relevant explanations of the equations. The revised parts in section 2.1 were indicated in red, including eq.1, eq.2, eq.4 and eq.5.

Point 5: In eq.2, Where is the dependency from t? what is x?

Response 5: We are very sorry for incorrectly writing the formula, which has been modified in the eq.2. t is the length of observation time in each data segment and x is the data points in each data segment. The revised parts in the manuscript were indicated in red.

Point 6: ‘For the selection of m, which must satisfy m>2d+1 in Takens theory, where d represents the real dimension of attractors.’ No meaning. To be rephrased.

Response 6: In this sentence, we want to denote that the original value of m should meet Takens theory (that is m>2d+1), which is a necessary condition. The formula m>2d+1 is a part of Takens theory. We have modified this sentence and changed the symbol ‘d’ to the symbol ‘h’, to avoid confusion with the symbol in the section 2.1. The revised parts in the section 2.2 were indicated in red. The revised parts in the section 2.2 were as follows:

‘For the selection of m in this paper, false nearest neighbors (FNN) [42] are introduced to obtain m, where m must satisfy m > 2h + 1 (Takens theory) [43] and h represents the real dimension of attractors.’

Point 7: It is unclear why the type of used wavelet shouldn't matter.

Response 7: We appreciate the professional suggestion of the reviewer. There is something unclear in our words. The selection of mother wavelet function is important, but scholars had done the relevant research on its selection, and come to the conclusion that coifN (N=5) wavelet function is more suitable for MT signal processing. Therefore, we quote the previous researches to solve the selection of the mother wavelet function. We have modified this sentence. The revised parts in the section 2.3 were indicated in red. The revised parts in the section 2.3 were as follows:

‘In MT signals, coifN is most suitable as the mother wavelet function, where N = 5 obtains a better result [45,46].’

Point 8: In Fig.3, the associated caption is incomplete. It is not clear what is what.

Response 8: We have modified this question. All the information in Fig.3 are explained in its caption, and we have modified Fig.3 to make readers more easily understand the meaning of Fig.3. The revised parts in the caption of Fig.3 were indicated in red. The revised parts in the caption of Fig.3 were as follows:

‘Figure 3. Entropy value of noise-free data and noisy data at different data segments. (a) Noise-free MT data, (b) data contaminated by square noise, (c) data contaminated by charge–discharge triangular wave, (d) data contaminated by impulse noise, (e) data contaminated by various noise. The green dashed line is the baseline of MDE, and the blue dashed line is the baseline of MSE. The red solid line shows the MDE, and the black solid line shows the MSE.’

Point 9: ‘In Table 1, it shows average entropy of the useful signal and noisy signals, to obtain the differences between various entropy values in the signals.’ to be rephrased.

Response 9: We have modified this question to express this sentence more clearly and correctly. The revised parts in the section 3.1 were indicated in red. The revised parts in the section 3.1 were as follows:

‘The MDE was calculated for the useful signal and noisy signal shown in Table 1 in order to test whether the difference between different signals could be found.’

Point 10: In Fig.5, what are the top rows?

Response 10: We appreciate the reviewer for reminding us of what we might neglect in the original manuscript. In Fig.5, the top rows are the processing results of noise-free data. We originally wanted to use the top rows to show that our method would not lose the useful signal, but the meaning was not clearly expressed, so the top rows are meaningless. We have modified this question with deleting the top rows. The revised parts in the Fig.5 were indicated in red.

Special thanks to you for your recognition.

We appreciate for Editors/Reviewers’ warm work earnestly, and hope that the correction will meet with approval.

Once again, thank you very much for your comments and suggestions.

Reviewer 4 Report

The authors have replied all my questions

Author Response

Response to Reviewer 4 Comments

Thank you for your affirmation of our manuscript, which gives us the motivation to research and write. This time we have made some revision to the paper, hoping to get your approval.

The revised parts in the manuscript were indicated in red.

Special thanks to you for your recognition.

We appreciate for Editors/Reviewers’ warm work earnestly, and hope that the correction will meet with approval.

Once again, thank you very much for your comments and suggestions.
